# The Difficulties in Interpersonal Regulation of Emotions Scale (DIRE): Factor Structure and Measurement Invariance across Gender and Two Chinese Youth Samples

**DOI:** 10.3390/bs14020125

**Published:** 2024-02-08

**Authors:** Yanhua H. Zhao, Lili Wang, Yuan Zhang, Jiahui Niu, Min Liao, Lei Zhang

**Affiliations:** 1School of Psychology, Henan University, Jinming Campus, Kaifeng 475001, China; yz@vip.henu.edu.cn (Y.H.Z.); wanglili@henu.edu.cn (L.W.); zhangy9212@henu.edu.cn (Y.Z.); njh@henu.edu.cn (J.N.); 104754230573@henu.edu.cn (M.L.); 2School of Computer and Information Engineering, Henan University, Jinming Campus, Kaifeng 475001, China

**Keywords:** difficulties in interpersonal emotion regulation, venting, excessive reassurance-seeking, difficulties in intrapersonal emotion regulation, well-being

## Abstract

Effective interpersonal emotion regulation (IER) strategies have been found to be meaningful predictors for positive psychological functioning. The Difficulties in Interpersonal Regulation of Emotions Scale (DIRE) is a measure developed to assess maladaptive IER strategies. This study aimed to examine the psychometric properties of the Chinese version of DIRE using two college student samples (Sample 1: *n* = 296; Sample 2: *n* = 419). The two-factor structure of DIRE (venting and excessive reassurance-seeking) was confirmed through an exploratory structure equation modeling approach. Our results demonstrated that the Chinese version of DIRE exhibits a similar factor structure (in both samples) as the original DIRE. Measurement invariance across gender and samples was also achieved. Latent mean analyses demonstrated that females more frequently reported excessive reassurance-seeking (in both samples) and venting (in Sample 1) than males. Furthermore, venting and excessive reassurance-seeking were significantly related to intrapersonal emotion regulation and well-being indicators. Although in Chinese culture DIRE performs somewhat differently from the original DIRE, the current findings suggest that DIRE is a reliable and valid scale with which to measure the IER strategies in Chinese culture and the use of this measure in clinical practice may allow for an accurate assessment of emotion regulation deficits in clients from other diverse cultures.

## 1. Introduction

The exertion of external resources to regulate one’s emotions is a ubiquitous phenomenon, even though most emotion regulation is a self-involved intrapersonal process. If a person seeks to change their emotional state through an interpersonal process, it is known as intrinsic interpersonal emotion regulation (IER) [1]. The social baseline theory highlights that proximity to other people may help save resources when regulating emotions [2]. Thus, intrinsic IER could help us manage distress in multiple ways. Recent studies have demonstrated that intrinsic IER is a meaningful psychological attribute that is important when applied to promoting well-being and mental health [3,4,5]. Therefore, the research on intrinsic IER would be a promising research area for understanding people’s well-being.

Existent studies have shown that the failed regulation of one’s emotions can increase the occurrence of various psychopathological symptoms and decrease people’s well-being across cultural groups [6,7]. In other words, whether through internal (intrapersonal emotion regulation) or external resources (intrinsic IER), failures in emotion regulation may decrease an individual’s well-being and increase psychopathological problems. The effects of different patterns of intrinsic IER on well-being outcomes have remained unclear until now [5]. Existent studies have revealed inconsistent results regarding the effects of intrinsic IER patterns. For example, Hofmann and research [4] developed a 20-item Interpersonal Emotion Regulation Questionnaire (IERQ), which identified four IER strategies including enhancing positive emotions, soothing, perspective taking, and social modeling in a general population. Using this measure, several studies [3,8] have presented inconsistent findings regarding the effects of soothing, social modeling, and perspective-taking on well-being and mental health. For example, soothing has been shown to increase depressive symptoms in general [9] but also decrease them under certain circumstances [10]. In a recent study [11], Messina et al. proposed a more comprehensive perspective on IER strategies, suggesting that excessive IER, indicative of difficulties in emotion self-regulation, may serve as a predictor of depression.

### 1.1. The Difficulties in Interpersonal Emotion Regulation Scale

Given the lack of extensive research, it is difficult to distinguish between adaptive and maladaptive patterns of intrinsic IER. For further investigation on intrinsic IER, several self-report measures have been developed, including the Interpersonal Emotion Regulation Scale [5] and the Difficulties in Interpersonal Regulation of Emotions Scale [12]. The DIRE was a self-report scale designed to measure maladaptive intrinsic IER strategies and highlighted the deliberate use of interpersonal resources to help regulate one’s emotions. This differs from one’s general regulation by other people. It measures two maladaptive IER strategies: venting (e.g., “raise voice or complain to the person in charge”) and excessive reassurance-seeking (e.g., “reassure themselves by keeping contacting friends or loved ones”). Two maladaptive intrapersonal emotion regulation (ER) strategies (acceptance and avoidance) were also included in this scale. The two intrinsic IER strategies and two intrapersonal ER strategies were embedded in three life scenarios: task-related, romantic, and social scenarios. The original validation study shows satisfactory internal consistency for each strategy on the scale. DIRE could assess IER strategies across various domains, which could be especially pertinent to psychopathology. In the original validation study [12], the frequent use of venting and excessive reassurance-seeking were associated with increased difficulties of intrapersonal ER and psychopathological symptoms. DIRE assesses maladaptive intrinsic IER and presents an emerging line of research. In this study, we are interested in the interpersonal strategies in the DIRE scale; thus, we use DIRE to represent the scale of difficulties of IER (venting and excessive-reassurance seeking) and attempt to examine the two-factor structure of DIRE in a different cultural context during the pandemic.

### 1.2. Association with Well-Being

IER has important applications in well-being. Using Hofmann’s IERQ, several studies have suggested that adaptive IER strategies are positively associated with life satisfaction and fewer symptoms of depression and anxiety. As for the maladaptive IER strategies, findings regarding the effects of venting and excessive reassurance-seeking on psychological functioning were inconsistent. Although the catharsis theory states that venting may help people feel better [13], many studies argued that venting can increase one’s aggressiveness and anger, intrapersonal ER difficulties, and almost all psychopathological symptoms [11,14].

As a maladaptive ER strategy, excessive reassurance-seeking is positively associated with interpersonal problems, the difficulties of intrapersonal ER, excessive soothing-seeking, and almost all psychopathological symptoms [15,16,17,18]. Reassurance-seeking as an intrinsic IER strategy can be difficult to differentiate from soothing-seeking. Several studies have shown that the frequent use of soothing to regulate emotions will decrease well-being [3]. Inconsistent with the above findings, excessive reassurance-seeking has been found to be positively associated with positive emotions and negatively associated with symptoms of depression [19,20]. Considering these inconsistent findings, we will further explore the effects of venting and excessive reassurance-seeking on well-being.

### 1.3. Emotion Regulation in Chinese Culture

Several differences in emotion regulation between Eastern and Western cultures have been identified in previous studies, reflecting culture-specific choices and a preference for strategies for emotion regulation [21,22,23]. For example, Chinese people tend to control emotions rather than express them, as opposed to those from the West [24,25]. Moreover, maladaptive strategies like expression suppression may have a different impact on the well-being of Eastern and Western culture samples [26]. Jobson and his colleges [23] demonstrated that the use of IER strategies may be different in Eastern culture, in which people are inclined to adopt IER (especially soothing) to improve emotion. Compared with intrapersonal ER, there are relatively few studies focusing on the adaptive and maladaptive strategies in IER and the cultural differences of IER [27]. Therefore, further studies in this area would bring new insights into understanding the different applications of IER in different cultures.

### 1.4. The Present Study

Given the current state of the measurement, DIRE is best for assessing the maladaptive strategies of intrinsic IER. Thus, this study examined the factor structure and measurement invariance of the DIRE as well as the latent gender differences of intrinsic IER strategies using two Chinese youth samples. First, we used an exploratory structural equation modeling approach (ESEM) to confirm the two-factor structure of the scale in two samples. Second, we investigated both configural, metric, and scalar measurement invariance of the best fitting models across the genders and samples. Finally, we examined correlations between the intrinsic IER strategies and an individual’s emotional expression, difficulties in intrapersonal ER, and well-being indicators. We expected individuals with higher scores on venting and excessive reassurance-seeking to self-report greater emotional expression and intrapersonal ER and lower well-being during the pandemic.

## 2. Materials and Methods

### 2.1. Participants and Procedure

To present the scientific inquiry on DIRE, this study used two independent samples. Sample 1 (*n* = 296; aged from 17–21, Mage = 18.30, SDage = 0.41; 65.5 % female; 96.3% Han Chinese, 3.7% minority) were all business major undergraduates from a comprehensive university in central China. Participants responded to a paper questionnaire that included demographic information and all the following measures. Participants were recruited through university classes and received a box of stationery as an incentive for their participation. Sample 2 (*n* = 419; 82.3% 15–25 years old, 15.5% 26–35 years old, 2.2% 36–45 years old; 54.2% females) were respondents from a broad educational background (49.7% arts, 41.5% sciences, and 8.8% others) and regions (currently enrolled college students nationwide were eligible to participate). The majority are undergraduates (81.8%) and master’s students (16.7%), with a minority pursuing doctoral degrees (1.4%). Among them, 51.6% were involved in a romantic relationship either previously or concurrently. They answered a short paper questionnaire including demographic information and the DIRE scale. Participants were recruited via advertisements on social media platforms and received an electronically generated red envelope (containing varying amounts of electronic money for online payments and shopping) upon completion of their online survey. Prior to questionnaire completion, all participants provided informed consent by signing consent forms. The studies involving human participants adhered to ethical standards approved by the local University Review Board.

### 2.2. Measures

*Difficulties in Interpersonal Emotion Regulation.* DIRE was used to measure the difficulties in both intrapersonal and interpersonal emotion regulation [12]. DIRE underwent a translation and back-translation process until all items were well understood. The interpersonal difficulties scale used here included two dimensions: venting (6 items) and excessive reassurance-seeking (6 items). Firstly, participants needed to evaluate their own status when facing three life scenarios, from 0 (not at all distressed) to 100 (extremely distressed). Then, they responded to the emotion regulation strategies on a scale of 1 (very unlikely) to 5 (very likely). The Cronbach’s alpha for venting was 0.75 (Sample 1) and 0.82 (Sample 2), and for excessive reassurance-seeking was 0.90 (Sample 1) and 0.91 (Sample 2).

*Difficulties in Intrapersonal Emotion Regulation.* Two scales were used to measure difficulties in intrapersonal ER. One scale was a revised intrapersonal ER difficulties scale selected from DIRE. The original intrapersonal ER difficulties scale included two dimensions: acceptance (3 items) and avoidance (6 items). Since previous studies suggest that acceptance may be a positive emotion regulation strategy in predicting psychopathological symptoms [28], we replicated the acceptance strategy for rumination (“Your feelings are your sole concern”). Rumination has been found as a maladaptive strategy for regulating one’s emotions (see review [28]). The rumination item was embedded into the DIRE across three scenarios as the system was designed for all items. Participants responded to this revised intrapersonal ER difficulty scale (including avoidance and rumination) from 1 (very unlikely) to 5 (very likely). The factor structure of this scale was confirmed by estimating several ESEM models with oblique rotation for exploratory purpose (using Sample 1, χ^2^ (*n* = 296) = 43.442, df = 17, *p* < 0.01, CFI = 0.958, TLI = 0.912, RMSEA = 0.073, SRMR = 0.034.) and target rotation for confirmatory purpose (using Sample 2, χ^2^ (*n* = 419) = 27.205, df = 17, *p* > 0.05, CFI = 0.979, TLI = 0.956, RMSEA = 0.038, SRMR = 0.031.), which resulted in a good model fit (factor loadings are available upon requirement). The Cronbach’s alpha for rumination was 0.76 (Sample 1) and 0.74 (Sample 2), and for avoidance it was 0.79 (Sample 1) and 0.69 (Sample 2).

Another scale was a brief Chinese version of the DIRE used to assess difficulties in intrapersonal ER [7]. This 15-item scale contains five difficulty domains: clarity (lack of emotional clarity), nonacceptance (nonacceptance of negative emotions), goals (difficulties engaging in goal-directed behaviors when distressed), impulse (difficulties controlling impulsive behaviors when distressed), and strategies (limited access to emotion regulation strategies perceived as effective). Participants rated their status corresponding with the description from 1 (almost never) to 5 (almost always). The Cronbach’s alpha for the total scale was 0.90, clarity was 0.77, nonacceptance was 0.80, goals was 0.85, impulse was 0.83, and strategies was 0.80.

*General Self-Efficacy.* A single-item general self-efficacy scale was used to measure perceived self-efficacy [29]. Participants responded to the item “I am confident in my ability to solve problems that I might face in life” on a 10-point scale from 1 (disagree strongly) to 10 (agree strongly).

*Socially Anxious Behavior.* Participants’ fear of social interaction was measured by the fear of social interaction subscale from the Chinese version of the Liebowitz Social Anxiety Scale [30]. The 11-item sale was rated from 1 (not at all) to 4 (very much). The Cronbach’s alpha for the scale was 0.85.

*Positive and Negative Affect.* The affect participants felt in the past week were measured by a Chinese version of the Positive and Negative Affect Scale [31]. The 12-item scale included two dimensions: positive affect (e.g., enthusiastic) and negative affect (e.g., scared). Participants rated their feelings on a 7-point scale from 1 (almost all time) to 7 (never). The Cronbach’s alpha for positive affect was 0.81 and for negative affect it was 0.71.

*Loneliness.* Participants’ feelings of loneliness was measured using the 5-item loneliness scale (UCLA-5) [32]. The scale was rated on a 4-point scale from 1 (never) to 4 (often). The Chinese version of the UCLA Loneliness Scale has been reported with good reliability and validity [33]. Since there is no report on the use of UCLA-5 in the Chinese sample, the one-factor structure model was examined by confirmatory factor analysis, which resulted in an acceptable model fit, χ^2^ = 6.863, df = 4, *p* = 0.143; CFI = 0.994, TLI = 0.985, RMSEA = 0.049(0.001–0.110), SRMR = 0.021. The Cronbach’s alpha for the scale was 0.82.

*Depressive Symptoms.* Participants’ depressive symptoms were measured by a 7-item subscale from the depression, anxiety, and stress scale [34]. Participants responded to the items on a 4-point scale from 0 (did not apply to me) to 3 (applied to me very much or most of the time). The Cronbach’s alpha for the scale was 0.86.

### 2.3. Data Analyses

To examine the structure of DIRE, exploratory structural equation modeling (ESEM) and multi-group analysis were conducted using Mplus 8.3. The remaining analyses were performed with SPSS 25. The following model fit indices were applied in this study: Comparative Fit Index (CFI), Tucker–Lewis Index (TLI), the Root-Mean-Square-Error of Approximation (RMSEA), and Standardized Root Mean Square Residual (SRMR). According to the cutoff value suggested by Hu and Bentler [35], CFI ≥ 0.90, RMSEA ≤ 0.08, and SRMR ≤ 0.10 indicate an adequate model fit. For model comparison, the change of CFI ≤ 0.01 and the change of RMSEA ≤ 0.015 indicates significant fit improvement [36].

## 3. Results

### 3.1. Factor Analysis

The proportion of missing values in Sample 1 was less than 1%, and no missing values were detected in Sample 2. The proposed two-factor ESEM model was estimated in Mplus 8.3 with a robust maximum likelihood (MLR) which did not assume the multivariate normality of data. Using Sample 1, data analysis provided a poor model fit for the two-factor model, χ^2^ (*n* = 296) = 255.973, df = 43, *p* < 0.05, CFI = 0.811, TLI = 0.709, RMSEA = 0.130 (0.115, 0.146), SRMR = 0.050. Following recommendations from the original validation study [12] and guided by modification indices from statistical analysis, we identified several notable residual covariances among items with similar wording or underlying latent constructs. These residual covariances were permitted to intercorrelate in the present study (see Figure 1). The model fit significantly changed, χ^2^ (*n* = 296) = 49.224, df = 34, *p* < 0.05, CFI = 0.986, TLI = 0.974, RMSEA = 0.039 (0.007, 0.062), SRMR = 0.027 (see Table 1). Subsequently, Sample 2 was employed to validate the factor structure derived from Sample 1. The results indicated that the model developed in Sample 1 exhibited good fit in Sample 2, as evidenced by the following indices: χ^2^ (*n* = 419) =123.48, df = 34, *p* < 0.001, CFI = 0.953, TLI = 0.909, RMSEA = 0.079 (0.064, 0.095), SRMR = 0.038 (see Table 1). Standardized item loadings of the ESEM model for both Sample 1 and Sample 2 are presented in Table 2.

### 3.2. Measurement Invariance and Gender Differences

The present study assumed that DIRE measured the same construct across genders and samples. Multigroup CFA analyses in two samples revealed that a model that constrained intercepts and loadings (metric and scalar invariance model) equal across gender groups was not significantly worse than the configural model. This suggested that strict measurement invariance across gender was achieved in both samples (see Table 1). Multigroup analyses examining the measurement invariance across samples revealed that the structure of DIRE is also equivalent across two samples (see Table 1).

**Table 1 behavsci-14-00125-t001:** Model fit.

Model	S-B Scaled χ^2^	df	CFI	TLI	RMSEA (90% CI)	SRMR	ΔCFI	ΔRMSEA
Sample 1 Two-factor ESEM model	49.224 *	34	0.986	0.974	0.039 (0.007, 0.062)	0.027		
Gender equivalence								
Configural	104.190 *	68	0.970	0.941	0.060 (0.035, 0.082)	0.041		
Metric	126.911 *	88	0.968	0.951	0.055 (0.032, 0.075)	0.061	−0.002	−0.005
Scalar	137.760 *	98	0.967	0.955	0.053 (0.030, 0.072)	0.064	−0.001	−0.002
Sample 2	123.480 *	34	0.953	0.909	0.079 (0.064, 0.095)	0.038		
Two-factor ESEM model								
Gender equivalence								
Configural	165.650 *	68	0.950	0.902	0.083 (0.067, 0.099)	0.042		
Metric	184.744 *	68	0.950	0.925	0.072 (0.058, 0.087)	0.057	<0.001	−0.011
Scalar	209.021 *	98	0.943	0.923	0.074 (0.060, 0.087)	0.060	−0.007	0.002
Measurement invariance across samples	135.514 *	34	0.967	0.935	0.078 (0.067, 0.089)	0.032		
Configural	177.887 *	68	0.964	0.930	0.067 (0.055, 0.080)	0.034		
Metric	198.565 *	88	0.964	0.946	0.059 (0.048, 0.070)	0.039	<0.001	−0.008
Scalar	211.997 *	98	0.963	0.950	0.057 (0.047, 0.068)	0.040	−0.001	−0.002

Note. S-B scaled χ^2^ = Satorra Bentler chi-square (weighted least square estimator was used); df = degrees of freedom; CFI = Comparative Fit Index; TLI = Tucker–Lewis Index; RMSEA = Root-Mean-Square Error of Approximation; 90% CI = 90% Confidence Interval for the RMSEA; ESEM = Exploratory Structural Equation Model; * *p* < 0.05.

**Table 2 behavsci-14-00125-t002:** Standardized factor loadings of DIRE for Sample 1 and Sample 2.

		Sample 1		Sample 2	
Scenario	Item	Vent	Reassure	Vent	Reassure
Task-related	Raise your voice or complain to the person in charge	0.73	−0.19	0.72	−0.13
Romantic	Raise your voice or complain to the person in charge	0.72	−0.11	0.75	−0.04
Social	Raise your voice or complain to the person in charge	0.51	0.03	0.40	0.22
Task-related	Complain to your coworkers or classmates about how unfair the situation is	0.54	0.00	0.63	0.01
Romantic	Complain to your coworkers or classmates about how unfair the situation is	0.42	0.24	0.73	−0.02
Social	Complain to your coworkers or classmates about how unfair the situation is	0.37	0.34	0.42	0.34
Task-related	Keep contacting friends and loved ones	0.12	0.40	0.18	0.48
Romantic	Keep contacting friends and loved ones	0.04	0.48	0.16	0.46
Social	Keep contacting friends and loved ones	−0.07	0.83	−0.16	0.91
Task-related	Keep asking for reassurance	0.17	0.60	0.22	0.58
Romantic	Keep asking for reassurance	0.07	0.65	0.12	0.71
Social	Keep asking for reassurance	−0.05	0.96	−0.11	0.99
Cronbach’s alpha		0.75	0.90	0.82	0.91
Latent correlation			Reassure		Reassure
Vent			0.49		0.61

After building an invariance across gender groups, the latent mean analysis was conducted to test the difference in DIRE between gender groups and samples. The latent values of males were set to zero in the strict invariance model, and the latent mean values of females were compared with males. Results indicated that, compared with males, females appeared to use excessive reassurance-seeking, and not venting, significantly more in Sample 1 (*p* > 0.05) and that they showed more intense venting and excessive reassurance-seeking in Sample 2 (see Table 3). Participants in Sample 2 reported slightly higher excessive reassurance-seeking than in Sample 1.

### 3.3. Association with Well-Being

To examine the convergent and discriminant validity of the two dimensions of DIRE, we examined the associations between dimensions of DIRE and intrapersonal ER difficulties and well-being indicators (see Table 4). Findings showed that the links between difficulties in interpersonal ER (venting and excessive reassurance-seeking) and difficulties in intrapersonal ER were all significantly positive except the link between excessive reassurance-seeking and avoidance. As for well-being, venting was negatively associated with self-efficacy and positive affect (marginal) but positively associated with socially anxious behavior, loneliness, negative affect, and depressive symptoms. Excessive reassurance-seeking was positively associated with positive affect and negatively associated with depressive symptoms.

## 4. Discussion

The current study provided initial support for the use of the DIRE in Chinese youth samples to assess the individual differences in difficulties in IER. The original two-factor structure of DIRE (venting and excessive reassurance-seeking) exhibited a good model fit in the current study across two Chinese samples. Measurement invariances across genders and samples were supported. Gender differences were identified, with females reporting a higher intrinsic IER usage. The scale showed adequate internal consistency. As expected, venting was positively associated with difficulties in intrapersonal ER and negatively associated with well-being. However, excessive reassurance-seeking was positively associated with positive emotions and negatively with depressive symptoms. The findings of this study suggest that DIRE may be an ideal research and practical tool with which to assess the strategies of IER in diverse cultures.

In Sample 1, the results of the two-factor ESEM model indicated a good fit, which was achieved by allowing for covariances between excessive reassurance-seeking item residuals. These residual correlations were also presented in the original study [12]. We replicated this two-factor model in the second sample. Moreover, the proposed model was strictly equivalent across genders and samples, supporting the latent mean level comparison between gender and sample groups [37].

As for gender differences, females more frequently reported the use of excessive reassurance-seeking in both samples and more frequently reported venting in Sample 1, suggesting that females prefer to use IER strategies to regulate their distress compared with males. This supports previous research which found that females are more likely to use external resources to regulate their negative emotions [8], and that males are more likely to suppress their emotions, particularly in Chinese society [26]. As for the sample difference, Sample 2 reported higher use of excessive reassurance-seeking, which may be due to differences in the recruitment methods of the two samples. Participants from Sample 2 were attracted by social media, and some participants in this sample may likely have higher excessive reassurance-seeking tendencies and look more for confirmation from external sources. The process of participation and interaction with other people may provide the chance for them to reassure their inner selves.

The convergent and discriminant validation of DIRE is assessed through DIRE’s associations with other self-reporting tools to measure individuals’ intrapersonal ER and well-being. There were moderate positive correlations between venting and difficulties in intrapersonal ER. This is consistent with previous studies [9,12,38] which found that intrinsic IER was positively correlated with intrapersonal emotion dysregulation. In line with previous studies [12,39], venting may increase negative emotions (loneliness, socially anxious behavior, and symptoms of depression) and decrease positive social and emotional well-being (self-efficacy and positive emotions). However, the correlation between venting and negative emotions was not as robust as observed in previous studies [11,12]. One potential explanation is that in cultures characterized by high levels of interpersonal dependence, such as China, recipients of complaints may tend to respond with more indirect negativity.

Consistent with some previous studies [12] but inconsistent with others [15,18], excessive reassurance-seeking shows a small positive correlation with positive emotions and a negative correlation with depressive symptoms. There are other similar findings that suggest that seeking help and reassurance may improve an individual’s depression state and promote positive affect [19,20]. If intrapersonal emotion regulation is considered alongside IER, IER strategies may serve as beneficial approaches for individuals with limited internal resources to regulate their emotions. Furthermore, in a culture that prioritizes interpersonal harmony, distinguishing between excessive and non-excessive reassurance-seeking is challenging. Therefore, based on the findings of the current study, we could not classify reassurance-seeking as a maladaptive IER strategy.

The current study has several limitations. First, since the two samples used were composed primarily of young people, findings may not be generalizable to groups from different ages and educational backgrounds. Future studies should examine the psychometric properties of the DIRE using more diverse samples. Second, Sample 1 was overrepresented by females; the current findings could be understood with consideration of this aspect. Third, we used two population samples in the present study. A future study could replicate or extend these findings in clinical samples to investigate the effects of IER on the development of psychopathology. Finally, the internal consistency of the avoidance scale was slightly below the recommended cutoff of 0.70. In future studies, researchers may want to take this aspect into account when considering the use of this intrapersonal emotion regulation scale.

In conclusion, this study can increase researchers’ and clinicians’ confidence and ability to reliably and validly assess IER strategies. DIRE may prove to be an effective tool for researchers and clinicians to assess IER difficulties in diverse populations. The future use of this scale will likely help researchers further investigate the effects of difficulties in IER and provide evidence for the importance of targeting IER in treatment.

## Figures and Tables

**Figure 1 behavsci-14-00125-f001:**
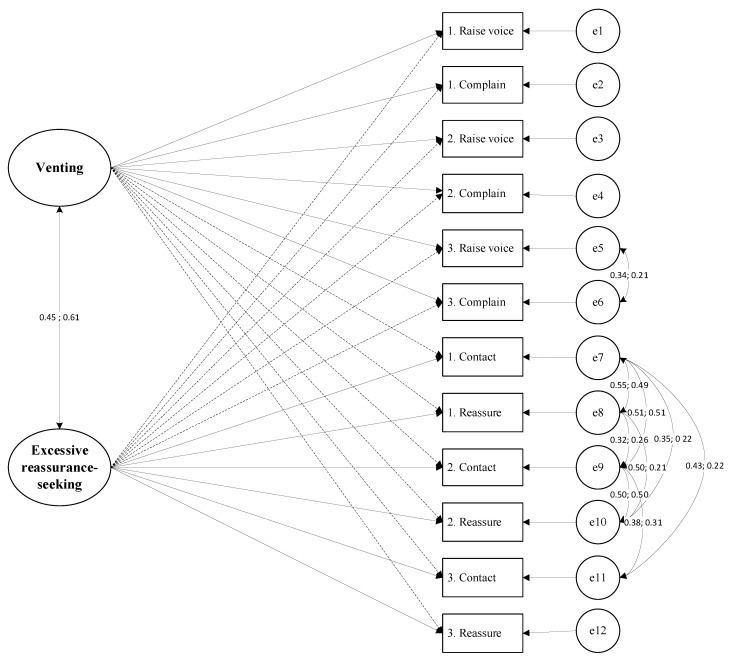
The two-factor Exploratory Structure Equation Model for Interpersonal Emotion Regulation Difficulties Scale in Sample 1 and Sample 2.

**Table 3 behavsci-14-00125-t003:** Estimated latent mean differences of DIRE factors between gender and sample group.

Factor	Males	Females	*p*	Pooled Standard Deviation	Cohen’s d	Vent	Reassure
Sample 1 (*n* = 296)	*n* = 102	*n* = 194				0	0
Vent	0	0.12	>0.05	1.08	0.11		
Reassure	0	0.48	<0.01	1.33	0.36		
Sample 2 (*n* = 419)	*n* = 192	*n* = 227				0.16	0.22
Vent	0	0.54	<0.01	1.32	0.41		
Reassure	0	0.27	<0.01	1.15	0.23		
*p*						>0.05	<0.05
Pooled standard deviation						1.1	1.13
Cohen’ d						0.15	0.19

**Table 4 behavsci-14-00125-t004:** Correlations between DIRE and other self-reported measures.

Variable	Vent	Reassure	Ruminate	Avoid
Intrapersonal Difficulties Scale (DIRE)				
1. Ruminate	0.49 **	0.14 *		
2. Avoid	0.30 **	0.07		
Difficulties of Emotion Regulation (ERD)				
1. Clarity	0.15 *	0.04	0.24 **	0.03
2. Nonacceptance	0.22 **	0.14 *	0.12 *	−0.01
3. Goals	0.24 **	0.13 *	0.23 **	0.07
4. Impulse	0.35 **	0.06	0.34 **	0.06
5. Strategies	0.27 **	0.08	0.32 **	0.01
6. ERD Total	0.32 **	0.12 *	0.33 **	0.05
Well-being (Sample 1)				
1. Self-efficacy	−0.15 *	−0.07	−0.24 **	−0.03
2. Socially anxious behavior	0.19 **	0.03	0.18 **	0.03
3. Loneliness	0.12 **	−0.08	0.24 **	0.11
4. Positive affect	−0.11 ^†^	0.14 *	−0.20 **	0.04
5. Negative affect	0.18 **	0.04	0.21 **	−0.00
6. Depressive symptoms	0.16 **	−0.12 *	0.28 **	0.10
7. Emotional distress (Sample 1/Sample 2)	0.23 **/33 **	0.08/26 **	0.18 */23 **	0.05/0.7

Note. ^†^
*p* < 0.06, * *p* < 0.05, ** *p* < 0.01.

## Data Availability

The datasets used and/or analyzed during the current study are available from the corresponding author upon reasonable request.

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
