# Peer review of "The Difficulties in Interpersonal Regulation of Emotions Scale (DIRE): Factor Structure and Measurement Invariance across Gender and Two Chinese Youth Samples"

_behavsci, 2024, doi:10.3390/bs14020125_

Round 1
Reviewer 1 Report
Comments and Suggestions for Authors
The discussion can be improved with a deep reflection of the results. Beyond cultural differences, other factors could be considered (for example, due to the sample, age effects may explain in part the results). Moreover, the combination of results coming from different measures may help in the interpretation of the results. For example, reassurance seek is negatively associated with depression, but positively with all DERS subscales.
I suggest also this new paper that could be cited: Messina, I., Maniglio, R., & Spataro, P. (2023). Attachment insecurity and depression: The mediating role of interpersonal emotion regulation. Cognitive Therapy and Research, 1-11.
Reviewer 2 Report
Comments and Suggestions for Authors
The manuscript presents an interesting study in which the psychometric properties of the scale Difficulties in interpersonal regulation of emotions in Two Chinese Youth Samples: Psychometric Properties are explored.
However, there are some issues that could clarify the manuscript. Specifically:
1. The title includes the "(...) and Association with Well-Being. However, this aspect is not central to the manuscript, but is used as a measure of convergent validity. Actually, the manuscript tries to find out the psychometric properties of the instrument in Chinese.
2. Similarly, the analysis of gender differences would also not be an essential part of the article. My recommendation is that it should not be included and, in any case, a subsequent study should be carried out, with a larger and more representative sample, in which some variables that may be differential in the difficulties in interpersonal regulation of emotions are studied.
3. The article in which the original questionnaire is found refers to "difficulties in interpersonal regulation of emotions (DIRE)", which coincides with other articles in which the instrument has been validated in another language. In no case does it appear with the abbreviation DIER.
4. On the other hand, in the original article in which the instrument is published, it has two parts, one interpersonal, the other intrapersonal, each with two factors, and with 21 items in total. However, in their manuscript only the interpersonal part is considered, and curiously the intrapersonal scale is used independently, for validity purposes, and with a validation study. Furthermore, as you describe, you have made some modifications to this scale. Wouldn't it have been more appropriate to validate the entire scale?
5. Although it is mentioned in the limitations, the results of the questionnaire cannot be extrapolated if it has been administered mainly to university students, 82% of whom are between 15 and 25 years old? And what about non-university students? Therefore, a more detailed description of both samples would be desirable. Location, universities involved, ... In short, a greater description of the socio-demographic characteristics of both samples would be desirable.
6. In measures, it would be convenient to provide more information on the psychometric properties of the original version of the questionnaire to be validated.
7. More information should be provided on the psychometric properties of the scales translated into Chinese, which are, after all, the ones that have been used. In some cases a lot of information is provided, in others very little. If there are no studies on the translated scales, this should be calculated with the sample of the studies themselves.
8. The Alpha coefficient is often provided as a measure of reliability. However, it has limitations when used with discrete measures, such as questionnaire items, especially if there are many of them and in large samples, overestimating the value. It would be more convenient to provide the McDonald's omega coefficient. Moreover, MPLUS calculates it.
9. In Table 4, the first two lines are confusing: "Intrapersonal emotion regulation difficulties (Sample 1)" and "Intrapersonal Difficulties Scale (DIER)" Why these two lines?
Reviewer 3 Report
Comments and Suggestions for Authors
This interesting article dealt with a Chinese version DIER scale administered to two groups of Chinese youth. The paper was crisply written and generally communicated well with the reader.
I acknowledge here that I am able to evaluate the use and discussion of statistical analyses only in a general way, as the techniques employed lie at the limits of my expertise.
Here are my comments.
1. Some small issues appeared in the paper that I think were probably due to language. For example, what is a "stationary gift" and what is a "random money bag" (these were discussed in lines 125 and 131). I assume that following the university's ethics procedures (line 133) involved obtaining informed consent from participants. This should probably be stated explicitly. In line 136, the word "suffered" should probably be changed to "underwent". The way in which samples are labeled in the Abstract is not the same as in the text (sample1 versus Sample 1). The sentence describing missing values in the two samples (line 206) is awkward and should be rephrased.
2. What is the role, in this research, of the "romantic experience" mentioned in line 127? What does this mean?
3. Several of the indicators of internal consistency are at the low end. For example on line 142, and again later line 160-1. This should be noted.
4. The description of results from line 213 onward is unclear. Perhaps it is too abbreviated and the authors need to take more time in clarifying it.
5. Why was there such a difference between the structural complexities of the two factors (Fig 1).
6. In Table 2, the source of questions listed multiple times needs to be indicated and explained.
7. I was pleased to see that correlations with other measures were included (Table 4). However, these are all quite weak with explanatory power in the range of 1% to 9%.
Comments on the Quality of English Language/
Round 2
Reviewer 2 Report
Comments and Suggestions for Authors
All suggestions have been taken into account